# Sex modulation of faces prediction error in the autistic brain

Adeline Lacroix [1] [✉], Sylvain Harquel[1,2], Martial Mermillod[1], Marta Garrido [3,4], Leonardo Barbosa[1,5], Laurent Vercueil[1], David Aleysson[1], Frédéric Dutheil[6], Klara Kovarski [7,8] & Marie Gomot[9]

Recent research suggests that autistic females may have superior socio-cognitive abilities compared to autistic males, potentially contributing to underdiagnosis in females. However, it remains unclear whether these differences arise from distinct neurophysiological functioning in autistic males and females. This study addresses this question by presenting 41 autistic and 48 non-autistic adults with a spatially filtered faces oddball paradigm. Analysis of event-related potentials from scalp electroencephalography reveal a neurophysiological profile in autistic females that fell between those of autistic males and non-autistic females, high-lighting sex differences in autism from the initial stages of face processing. This finding underscores the urgent need to explore neurophysiological sex differences in autism and encourages efforts toward a better comprehension of compensation mechanism and a clearer definition of what is meant by camouflaging.

[1] Univ. Grenoble Alpes, Univ. Savoie Mont Blanc, CNRS, LPNC, 38000 Grenoble, France. [2] Defitech Chair in Clinical Neuroengineering, Center for Neuroprosthetics and Brain Mind Institute, EPFL, Geneva, Switzerland. [3] Cognitive Neuroscience and Computational Psychiatry Lab, Melbourne School of Psychological Sciences, University of Melbourne, Melbourne, VIC, Australia. [4] Graeme Clark Institute for Biomedical Engineering, University of Melbourne, Melbourne, VIC, Australia. [5] Fralin Biomedical Research Institute at VTC, Virginia Tech, Roanoke, VA 24016, USA. [6] Université Clermont Auvergne, CNRS, LaPSCo, CHU Clermont-Ferrand, WittyFit, F-63000 Clermont-Ferrand, France. [7] Sorbonne Université, Faculté des Lettres, INSPE, Paris, France. [8] LaPsyDÉ, Université Paris-Cité, CNRS, Paris, France. [9] UMR 1253 iBrain, Université de Tours, Inserm, Tours, France. [✉]email: adeline.lacroix@univ-grenoble-alpes.fr

Autism Spectrum Disorder (ASD), referred to as autism in respect of autistic individuals' preferences[1–3], is characterized by challenges in social communication, sensory specificities as well as stereotyped and repetitive behaviors. Despite temporal and geographic variation, the prevalence is estimated to be around 1% of the world's population[4–6]. The male-to-female ratio varies from 4:1 in children[5–7] to 2.6:1 in adults (according to a study based on the Norwegian Medical Birth Registry[5]). Missed or delayed early diagnosis of autistic females possibly due to sex differences in the autism phenotype, may be an underlying factor in this difference. Indeed, there is increasing evidence that autistic females exhibit stronger social skills or more normalized behavioral responses to social stimuli compared to autistic males[8–15]. However, it is unclear whether these behavioral differences between autistic males and females are also observed at the neurophysiological level.

Therefore, the present electroencephalography (EEG) study aimed to compare the neurophysiological response of autistic males and females as well as non-autistic (NA) adults during the initial stages of face processing. Indeed, face processing is an important part of socio-communicative abilities and is characterized by qualitative and quantitative differences in autism compared to NA individuals[16]. For example, autistic individuals often recognize emotions from faces less accurately and show slower responses than NA individuals[17,18]. fMRI studies show that the regions recruited during face processing differ between autistic and NA individuals[19] or that some regions such as amygdala or superior frontal gyrus are less activated[20,21]. In addition, a review of EEG studies highlighted a reduced amplitude and longer latency for the N170 response to faces in autism[22]. Despite sex differences in socio-communicative abilities in autism[15], sex differences in face processing have rarely been investigated, especially in neuroimagery. One study showed an attenuated N170 response in autistic girls compared to boys[23] but included only 24 autistic children and no control group.

The peculiarities in face processing could be partly explained by reduced global processing and/ or enhanced local processing in autism[24,25]. Indeed, autistic individuals would rely on High Spatial Frequencies of the images (HSF, conveying local information) more[26,27] or earlier[28,29] compared to NA individuals, including during face processing[30], but refs. [30–32]. On the contrary, the early and fast extraction by the primary visual cortex of Low Spatial Frequencies (LSF), conveying coarse information, would be the default processing in NA as it would help to make predictions, as suggested by Bar's model[33] (Fig. 1). More specifically, after extraction, the LSF would activate the orbitofrontal

cortex via the magnocellular pathway, where it would be used to generate predictions that would then be projected top-down to the primary visual cortex and inferotemporal areas to guide the subsequent integration of HSF[33–36]. According to the predictive brain framework, the brain's ability to generate predictions is involved through feedback connections to facilitate perception and learning[37–39]. Accurate predictions of faces and emotions may also be beneficial for navigating social environments[40,41]. Thus, we hypothesize that the autistic bias toward HSF may contribute to their difficulties in face processing by reducing their predictive processes from LSF. Nevertheless, the better social skills and attention to faces in autistic females compared to autistic males could be explained by more typical predictive processes in autistic females.

To investigate these questions, we used a controlled mismatch response (MMR) paradigm (see Methods section) similar to a previous study[42]. With MMR, the automatic detection of unpredicted events within a learned regularity can be analyzed[43–47] using scalp EEG. The most widely accepted theory interprets the greater MMR following an unpredicted stimulus, compared to the predictable repeated standard stimulus, as the neural correlate of prediction error[48–50]. Visual MMR has been observed in response to a variety of stimuli, including the detection of emotional changes in faces[51–53]. MMR has been found to be atypical in autism[54,55], consistent with theoretical frameworks suggesting predictive coding specificities in autistic individuals[54,56–58].

During the task, a neutral unfiltered face (i.e., containing the broad spectrum of spatial frequencies) was repeatedly presented as the standard stimulus, while the same face filtered in LSF (i.e., containing only LSF information) or HSF (i.e., containing only HSF information) was presented as deviants. In our previous study, we highlighted a reduced LSF MMR (i.e., MMR to deviants containing only LSF) compared to HSF MMR (i.e., MMR to deviants containing only HSF) in NA[42]. This confirmed that deviants containing only LSF information from the standard stimulus led to fewer prediction errors than deviants containing only HSF information. This can be interpreted as HSF deviants matching less strongly with predictions based on LSF than LSF deviants. This finding is consistent with LSF information being at the root of visual prediction processes during face processing.

The present study replicates previous findings with a larger sample of NA individuals. In addition, our hypothesis posited that reduced LSF predictions in autistic individuals would result in a smaller discrepancy between their LSF MMR and HSF MMR, relative to the NA group. This hypothesis was confirmed, supporting the validity of predictive brain atypicalities in autism. We also hypothesized that the neurophysiological profile of autistic females would be more typical compared to autistic males, a hypothesis that was supported by the results. The study demonstrated that the neurophysiological response of autistic women during automatic face processing was intermediate between that of autistic men and NA individuals. This may partly explain the increased socio-communicative abilities often observed in autistic females compared to males.

## Results

41 autistic adults (20 females) and 48 NA adults (24 females) participated in the MMR paradigm (see Methods section). An initial exploratory whole-brain analysis was performed across the entire time window (0 to 600 ms) using cluster-based permutation. Subsequent analyses focused specifically on the evoked components associated with the MMR in time and space (130–230 and 350–450 ms, in parieto-occipital and central areas) and the sensory responses to faces (P100, N170). For the MMR,

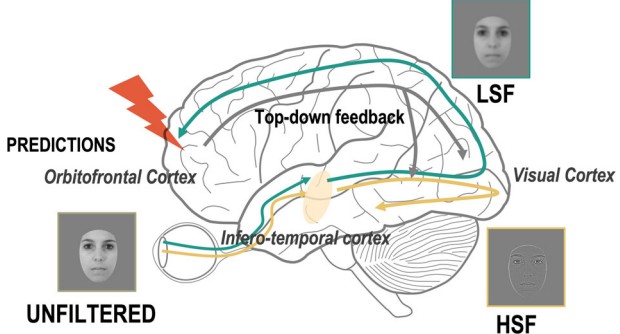

**Fig. 1 Bar's model applied to face perception.** Visual perception first relies on predictions from Low Spatial Frequencies. Fast extraction of Low Spatial Frequencies (LSF) and their projection onto the orbitofrontal cortex, through the dorsal visual stream (green arrow), enables predictions. Later integration of HSF through the ventral visual stream (yellow arrow) enables identification. Predictions may be reduced in autism (red flash).

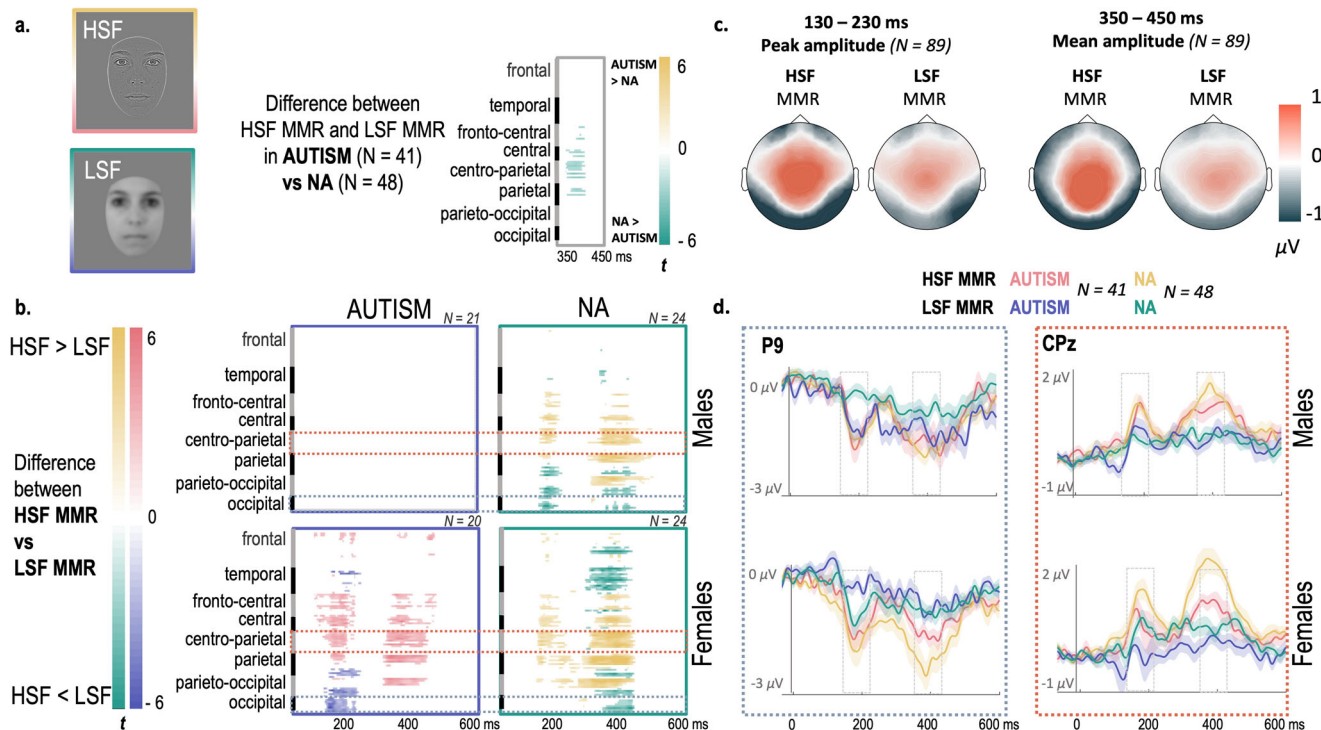

**Fig. 2 Modulation of the visual Mismatch Response (MMR) to faces containing only HSF (HSF MMR) and faces containing only LSF (LSF MMR) in respect to group (AUTISM and NA) and sex (FEMALES and MALES). a** Cluster statistics show a significantly reduced difference between HSF MMR and LSF MMR in autism compared to NA in the 350–450 ms time window over central, centro-parietal and parietal areas. **b** Cluster statistics show brain areas and time windows where HSF MMR is significantly larger than the LSF MMR, for each group and sex. The red and blue dotted rectangles indicate the areas where waveform analyses were performed (see **d**). **c** Topographies represent the mean MMR activity for each condition (HSF MMR and LSF MMR) in each time-window of interest (where the analyses were performed : 130–230 ms and 350–450 ms). **d** Waveforms represent the grand average visual MMR with standard error (shaded areas) at P9 and CPz for each group and sex. The gray rectangles represent the two time windows for which the analyses were performed.

analyses were performed on peak amplitude and latency in the first time window and on mean amplitude in the second time window (see Methods section). The detailed results of each model can be found in the Supplementary Materials (Supplementary Tables 1 through 11), along with supplemental correlation analyses (Supplementary Fig. 1 through 4). For each statistical model, planned comparisons were performed to decompose the interaction between Spatial Frequency (SF), Group and Sex to investigate sex differences in neurophysiological profiles[59,60].

**Spatial frequencies modulate the mismatch response differently in autism versus NA.** In NA, cluster-based permutation revealed a reduced difference in LSF MMR (i.e., MMR to deviants containing only LSF) compared to HSF MMR (i.e., MMR to deviants containing only HSF) over centro-parietal areas from 147 to 545 ms ($p_{cluster} = 0.002$), and over temporo-parietal and occipital regions from 158 to 251 ms ($p_{cluster} = 0.02$), and from 313 to 511 ms ($p_{cluster} = 0.002$).

In autism, there were two significant clusters indicating a reduced difference in LSF MMR compared to HSF MMR over centro-parietal areas from 108 to 231 ms ($p_{cluster} = 0.006$), and from 315 to 486 ms ($p_{cluster} = 0.002$). However, over temporo-parietal and occipital regions, in contrast to NA, there was a significant cluster only in the first time window, from 111 to 244 ms ($p_{cluster} = 0.008$), while no significant difference was found in the second time window.

This difference between NA and autistic individuals was confirmed with the between-group comparison in the two latency ranges of interest, in which cluster-based permutations revealed a reduced difference between HSF MMR and LSF MMR in autism

compared to NA between 367 and 411 ms post-stimulus ($p_{cluster} = 0.03$), as shown in Fig. 2a.

**Mismatch response specificity in autism is modulated by sex.** Cluster-based analysis in each subgroup revealed reduced LSF MMR compared to HSF MMR in autistic females, NA males and NA females on both time windows (all $p_{cluster} < 0.04$). In contrast, no significant difference was found in autistic males. These results are represented in Fig. 2b where colored areas show significantly larger HSF MMR than LSF MMR according to time and brain areas, observed over posterior (negative MMR) and central (positive MMR) areas. Sex differences depending on the group were further investigated with evoked potentials. Topographies (Fig. 2c) and potential waveforms (Fig. 2d) show a more strongly identifiable MMR in the HSF condition versus the LSF condition, in both groups (autism and NA) and for both sexes. In line with cluster-based statistics, visual inspection of MMR waveforms over posterior and central electrodes shows a first peak between 130 and 230 ms, representing the first time window of the MMR. There is also subsequent and more sustained activity between 350 and 450 ms, representing the second time window of the MMR.

Analyses of the first peak amplitude on parieto-occipital areas (Supplementary Table 2) showed a larger HSF MMR compared to LSF MMR with a large effect size ($F$ (1, 85) = 62.72, $p < 0.001$, $\eta_p^2 = 0.42$). We also found two significant interactions, between Sex and SF ($F$ (1, 85) = 8.09, $p = 0.006$, $\eta_p^2 = 0.09$), and between Group and Sex ($F$ (1, 85) = 3.99, $p = 0.049$, $\eta_p^2 = 0.04$), with no significant post-hoc paired comparisons. The interaction between Group, Sex and SF was not significant ($F$ (1, 85) = 0.42, $p = 0.520$,

$\eta_p^2 = 0.00$). However, planned comparisons revealed larger HSF MMR than LSF MMR in NA females ($p = {}< 0.001$, $\eta_p^2 = 0.27$), autistic females ($p = {}< 0.001$, $\eta_p^2 = 0.23$), and NA males ($p = 0.024$, $\eta_p^2 = 0.12$) but not in autistic males ($p = 0.624$, $\eta_p^2 = 0.04$). Peak latency analyses (Supplementary Table 3) revealed faster LSF MMR than HSF MMR ($F (1, 85) = 5.89$, $p = 0.017$, $\eta_p^2 = 0.06$), qualified by an interaction between Group, Sex, and SF ($F (1, 85) = 4.08$, $p = 0.047$, $\eta_p^2 = 0.05$), with no significant post-hoc paired comparisons. In the second time window, larger mean amplitude of HSF MMR compared to LSF MMR (found as a main effect: $F (1, 85) = 40.19$, $p < 0.001$, $\eta_p^2 = 0.32$; Supplementary Table 4), was observed in NA females ($p = 0.001$, $\eta_p^2 = 0.18$) and NA males ($p = 0.005$, $\eta_p^2 = 0.15$). The difference was not significant for autistic females ($p = 0.323$, $\eta_p^2 = 0.06$) and autistic males ($p = 0.279$, $\eta_p^2 = 0.06$). The interactions between Group and SF ($F (1, 85) = 2.37$, $p = 0.127$, $\eta_p^2 = 0.03$) and Group, Sex and SF ($F (1, 85) = 0.06$, $p = 0.810$, $\eta_p^2 = 0.00$) were not significant though.

Similar to the posterior responses, analyses on central areas of the peak amplitude (Supplementary Table 5) showed a larger HSF MMR compared to LSF MMR ($F (1, 85) = 41.88$, $p < 0.001$, $\eta_p^2 = 0.33$), qualified by an interaction between SF and Sex ($F (1, 85) = 4.87$, $p = 0.030$, $\eta_p^2 = 0.05$), but not between Group, Sex and SF ($F (1, 85) = 0.00$, $p = 0.968$, $\eta_p^2 = 0.00$). Indeed, the larger HSF MMR compared to LSF MMR appeared in both autistic females ($p = 0.007$, $\eta_p^2 = 0.14$) and NA females ($p < 0.001$, $\eta_p^2 = 0.22$), but not in autistic males ($p = 0.714$, $\eta_p^2 = 0.03$) and NA males ($p = 0.146$, $\eta_p^2 = 0.08$). There was no significant effect on latency (Supplementary Table 6). In the second time window, larger mean amplitude of HSF MMR compared to LSF MMR (found as a main effect : $F (1, 85) = 58.17$, $p < 0.001$, $\eta_p^2 = 0.41$; Supplementary Table 7), was observed in NA females ($p < 0.001$, $\eta_p^2 = 0.28$), NA males ($p = 0.004$, $\eta_p^2 = 0.15$), and in autistic females ($p = 0.016$, $\eta_p^2 = 0.13$), but not in autistic males ($p = 0.364$, $\eta_p^2 = 0.05$). The interaction between Group, Sex and SF was not significant though ($F (1, 85) = 0.04$, $p = 0.850$, $\eta_p^2 = 0.00$).

**Atypical source activity in autism is modulated by sex.** The source reconstructions for the HSF MMR and the LSF MMR for each group and each sex are shown in Fig. 3. The figure also represents the results of the statistical analysis for the comparison between sources of the HSF MMR filtered faces and the LSF MMR filtered faces in the two time windows for each subgroup. In the first time window, the results were in line with, and enhanced the cluster and waveform analyses. Indeed, autistic males did not display any area with larger HSF MMR activity compared to LSF MMR activity. On the contrary, in autistic females, larger HSF MMR activity compared to LSF MMR activity was found in two clusters in the right ($p_{cluster} = 0.017$) and left hemisphere ($p_{cluster} = 0.020$), including the fusiform area, infero-temporal cortex, isthmus cingulate and parahippocampal cortex. In NA females, the difference was located in the right fusiform gyrus as well as inferotemporal and middle temporal cortices ($p_{cluster} = 0.038$), and in NA males in the left temporal pole and entorhinal cortex as well as in the anterior parts of the left fusiform and middle temporal gyrus ($p_{cluster} = 0.038$). In the second time window, greater activity for the HSF MMR compared to the LSF MMR was observed only in NA females, with two significant clusters in the right ($p_{cluster} = 0.025$) and left

hemisphere ($p_{cluster} = 0.002$) including the left fusiform area, parahippocampal cortex, isthmus cingulate, and lateral part of the lateral-occipital cortex as well as the right isthmus cingulate.

**Reduced spatial frequency differentiation in autism is modulated by sex.** Analysis of the sensory response to standard and deviant stimuli is also critical in assessing differentiation (or lack thereof) in spatial frequency processing. These sensory responses are shown in Fig. 4 (for each group and sex). For the statistical analyses, we used the stimuli of the equiprobable sequence, i.e., the sequence in which each stimulus is presented with the same probability of occurrence, which served as a control (see Methods section).

The results (Supplementary Table 8) showed a large effect of SF on P100 ($F (1, 118) = 40.30$, $p < 0.001$, $\eta_p^2 = 0.32$) and on N170 ($F (1, 110) = 47.05$, $p < 0.001$, $\eta_p^2 = 0.36$). More specifically, the P100 after the Broad Spatial Frequency stimulus (i.e., the unfiltered stimulus containing the full spectrum of spatial frequencies, hereafter referred to as BSF P100) was larger than the LSF P100 (i.e., P100 after the stimulus containing only LSF; $p = 0.024$, $\eta_p^2 = 0.04$). The LSF P100 was also larger than the HSF P100 (i.e., P100 after the stimulus containing only HSF; $p < 0.001$, $\eta_p^2 = 0.18$). In addition, females presented a larger P100 to faces than males ($F (1, 85) = 8.88$, $p = 0.004$, $\eta_p^2 = 0.09$) but the interaction between Group, Sex and SF was not significant ($F (1, 118) = 0.93$, $p = 0.366$, $\eta_p^2 = 0.01$). Planned comparisons showed that the differences between spatial frequencies were observed in NA females (BSF P100 vs HSF P100 : $p < 0.001$, $\eta_p^2 = 0.14$; LSF P100 vs HSF P100 : $p = 0.008$, $\eta_p^2 = 0.08$) and NA males (BSF P100 vs HSF P100 : $p < 0.001$, $\eta_p^2 = 0.16$; LSF P100 vs HSF P100 : $p = 0.001$, $\eta_p^2 = 0.11$), whereas only the BSF P100 was larger than the HSF P100 in autistic females (BSF P100 vs HSF P100 : $p = 0.001$, $\eta_p^2 = 0.10$; LSF P100 vs HSF P100 : $p = 0.534$, $\eta_p^2 = 0.03$) and no significant difference was observed in autistic males (BSF P100 vs HSF P100 : $p = 0.550$, $\eta_p^2 = 0.03$; LSF P100 vs HSF P100 : $p = 0.843$, $\eta_p^2 = 0.02$).

While the LSF P100 (and BSF P100) was larger than the HSF P100, it also took longer to peak, as revealed by the analyses on P100 latencies (main effect of SF : $F (2, 130) = 36.36$, $p < 0.001$, $\eta_p^2 = 0.30$ - Supplementary Table 9; HSF vs LSF : $p < 0.001$, $\eta_p^2 = 0.28$; HSF vs BSF : $p < 0.001$, $\eta_p^2 = 0.18$). This effect, observed for NA females (HSF vs LSF : $p = 0.001$, $\eta_p^2 = 0.10$; HSF vs BSF : $p = 0.034$, $\eta_p^2 = 0.07$) and NA males (HSF vs LSF : $p < 0.001$, $\eta_p^2 = 0.16$; HSF vs BSF : $p < 0.001$, $\eta_p^2 = 0.12$), was observed only between HSF and LSF in autistic females (HSF vs LSF : $p = 0.045$, $\eta_p^2 = 0.06$; HSF vs BSF : $p = 0.085$, $\eta_p^2 = 0.05$) and was not observed in autistic males (HSF vs LSF : $p = 0.094$, $\eta_p^2 = 0.05$; HSF vs BSF : $p = 0.996$, $\eta_p^2 = 0.01$). The interaction between Group, Sex and SF was not significant though ($F (2, 130) = 1.62$, $p\ 0.207$, $\eta_p^2 = 0.02$).

Conversely, for the later N170 component responding more specifically to face, HSF N170 was larger than LSF N170 ($p < 0.001$, $\eta_p^2 = 0.18$) which was also larger than BSF N170 ($p = 0.003$, $\eta_p^2 = 0.06$). There was no interaction effect between Group, Sex and SF ($F (1, 110) = 0.16$, $p\ 0.754$, $\eta_p^2 = 0.00$; Supplementary Table 10), but planned comparisons showed these effects in NA females (HSF vs. LSF : $p < 0.001$, $\eta_p^2 = 0.11$; HSF vs. BSF : $p < 0.001$, $\eta_p^2 = 0.22$) and in NA males (HSF vs. LSF : $p = 0.008$, $\eta_p^2 = 0.08$; HSF vs. BSF : $p < 0.001$, $\eta_p^2 = 0.15$) whereas

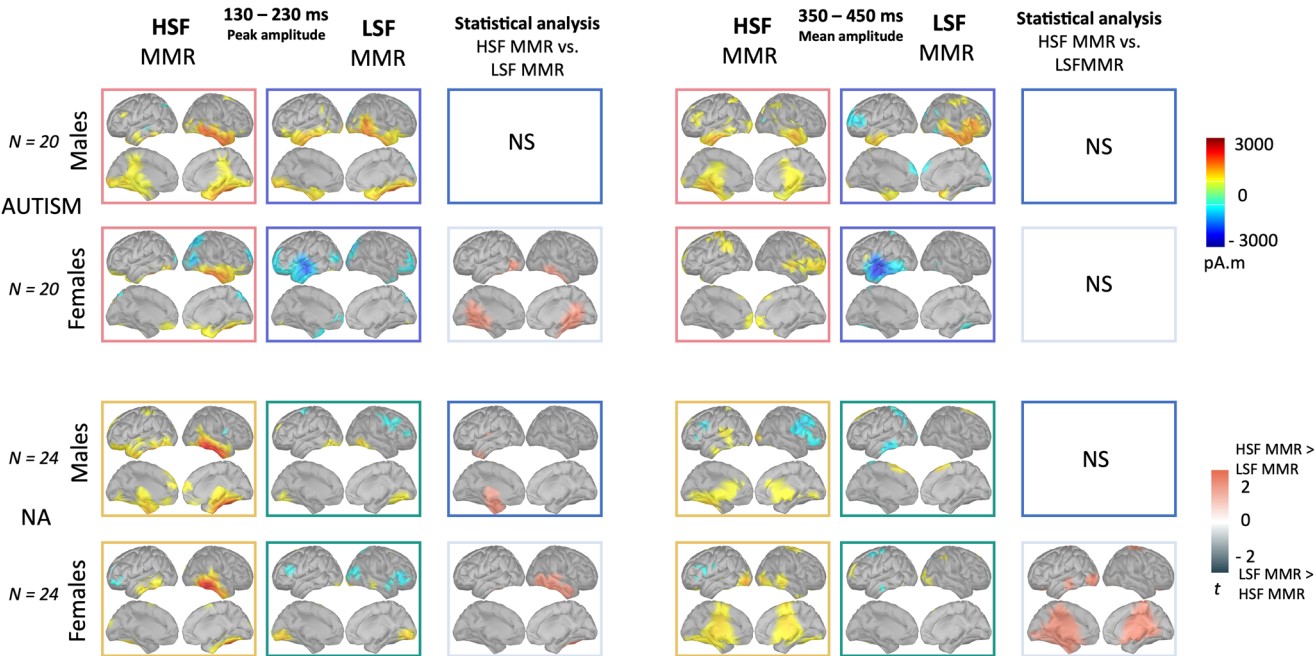

**Fig. 3 Modulation of source activity for each group (AUTISM and NA), sex (FEMALES and MALES) and condition (HSF MMR and LSF MMR).** Source activity (in pA.m) is presented for each time window (130–230 ms and 350–450 ms). Statistical analysis showing the difference in sources between conditions for each subgroup is also presented. NS non significant.

autistic females presented only a larger HSF N170 than BSF N170 (HSF vs. LSF : $p = 0.213$, $\eta_p^2 = 0.04$; HSF vs. BSF : $p < 0.001$, $\eta_p^2 = 0.11$) and autistic males presented no significant difference between conditions (HSF vs. LSF : $p = 0.979$, $\eta_p^2 = 0.01$; HSF vs. BSF : $p = 0.478$, $\eta_p^2 = 0.03$). Meanwhile, HSF N170 also peaked later (main effect of SF : $F (2, 167) = 23.64$, $p < 0.001$, $\eta_p^2 = 0.22$- Supplementary Table 11; HSF vs. LSF : $p < 0.001$, $\eta_p^2 = 0.16$; HSF vs BSF : $p < 0.001$, $\eta_p^2 = 0.19$) but the interaction between Group, Sex and SF was not significant ($F (2, 167) = 1.19$, $p = 0.307$, $\eta_p^2 = 0.01$). Planned comparisons revealed effects only for autistic males (HSF vs. LSF : $p = 0.017$, $\eta_p^2 = 0.07$; HSF vs. BSF : $p < 0.001$, $\eta_p^2 = 0.12$) but differences were not significant for autistic females (HSF vs. LSF : $p = 0.781$, $\eta_p^2 = 0.02$; HSF vs. BSF : $p = 0.937$, $\eta_p^2 = 0.01$), NA males (HSF vs. LSF : $p = 0.215$, $\eta_p^2 = 0.04$; HSF vs. BSF : $p = 0.053$, $\eta_p^2 = 0.06$) and NA females (HSF vs. LSF : $p = 0.132$, $\eta_p^2 = 0.05$; HSF vs. BSF : $p = 0.140$, $\eta_p^2 = 0.05$).

## Discussion

The objective of this study was to determine whether the EEG responses of autistic males and females are similar or different during the early stages of face processing. A controlled MMR paradigm was used. Results showed an intermediate neurophysiological response of autistic females, at a level between autistic males and non-autistic individuals, arguing in favor of sex differences in the neurophysiological signature of autistic males and females during automatic face processing.

The controlled MMR paradigm used in this study was specifically designed to investigate pre-attentive processes and predictive mechanisms to faces[42]. It used an unfiltered face as the standard and the same face filtered in LSF or HSF as the deviant. Relying on the coarse-to-fine processing of the visual perception framework (Fig. 1)[33,61], we expected a larger HSF MMR compared to LSF MMR in NA individuals. Indeed, HSF deviants

would not match LSF-based predictions from the standard stimulus and would, therefore, induce a larger prediction error. This was verified by our findings (Figs. 2 and 3), which are consistent with the coarse-to-fine processing of faces[13], possibly critical for rapid face processing and subsequent social adaptation. However, as expected in autism, cluster statistics did not reveal significant difference between HSF MMR and LSF MMR (in the second time window), in contrast to NA. This result is in line with decreased predictive processes from LSF during face processing in autism, but stratification by sex is needed to discuss this further.

The absence of a significant difference between HSF MMR and LSF MMR was observed throughout all analyses in autistic males (both in sensor and source space, see Figs. 2 and 3, respectively), in contrast to what is observed in the other groups. This may be due to decreased specialized processing between HSF and LSF in autistic males, as suggested by the lack of difference between HSF and LSF on the sensory response (P100 and N170; Fig. 4) in this group, extending previous findings in autistic children (mainly males[62,63]) to adults. These results corroborate the idea that, in contrast to NA, predictions during automatic face processing in autistic males would be less likely to rely on LSF, which may partly contribute to their difficulties in processing and anticipating facial information. This is an important result because it adds empirical data to support the predictive coding hypothesis, which has recently been proposed to explain autism characteristics[56–58], and substantiates that predictive coding specificities might contribute to social difficulties in autistic males.

Autistic females, similar to NA males and females, presented a larger negative HSF MMR compared to LSF MMR in the first time window (Fig. 2). This unimpaired early prediction error to spatial frequency changes suggests intact early predictive processes from LSF. This may be related to better spatial frequency differentiation compared to autistic males as autistic females have a reduced P100 to HSF compared to BSF (which includes LSF), similar to NA. This may indicate differences between autistic males and females in automatic low-level visual processing, and thus in sensory or perceptual functioning, which remain to be explored.

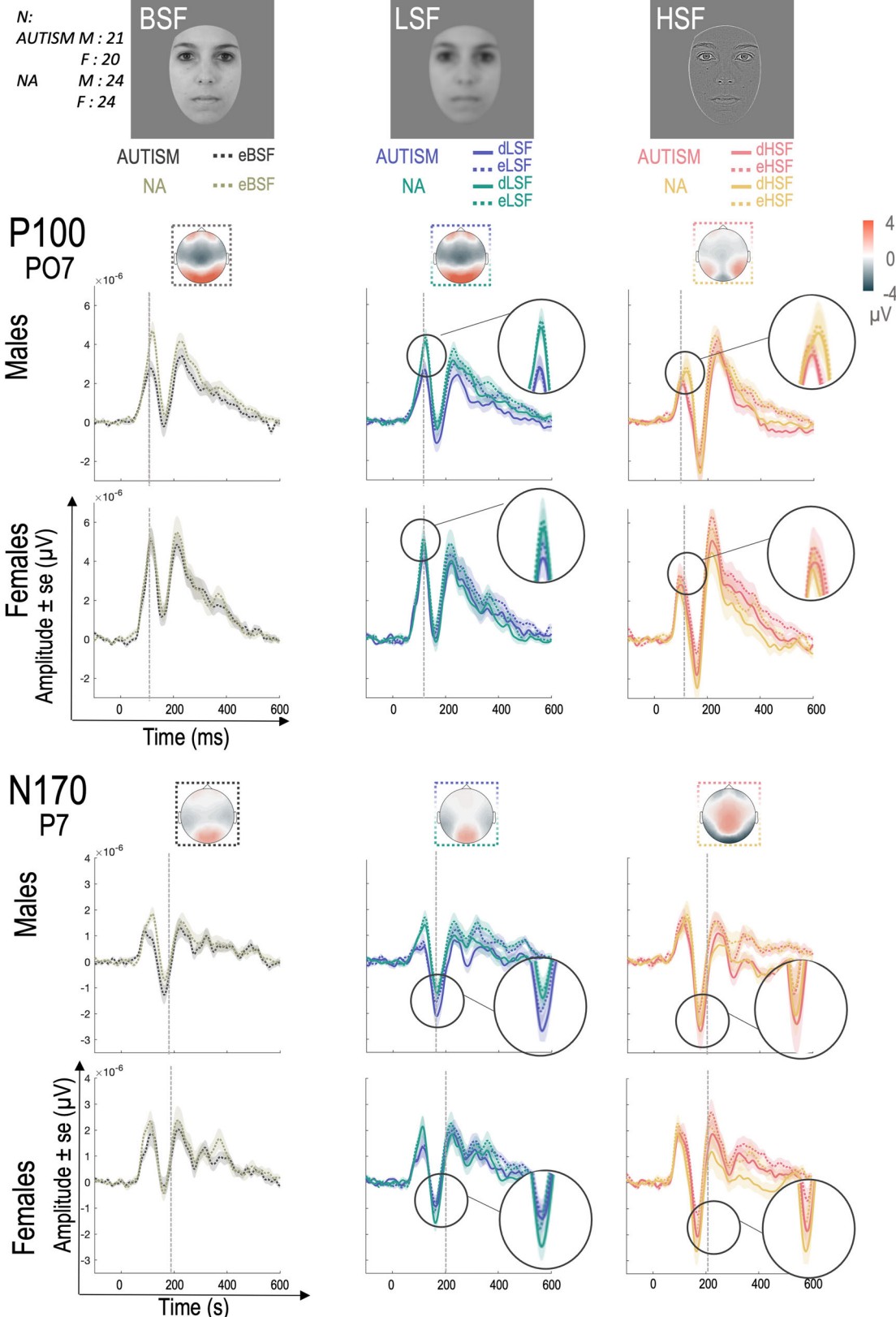

**Fig. 4 Modulation of the sensory response to faces (P100 on electrode PO7 and N170 on electrode P7) with respect to each group (AUTISM and NA), sex (FEMALES and MALES) and condition (Broad Spatial Frequencies—BSF, Low Spatial Frequencies—LSF and High Spatial Frequencies—HSF).** On each plot, the response to stimuli in the equiprobable sequence (eBSF, eLSF and eHSF; sequence in which stimuli appear with equal probability) is shown in dashed lines and the response to stimuli in the oddball sequence (dLSF for LSF deviants and dHSF for HSF deviants) is shown in solid lines. The shaded area indicates the standard error. Statistical analyses were performed on PO7 and PO8 for the P100 and P7 and P8 for the N170.

In addition to sex differences observed solely in autism, it can be hypothesized that some of the specificities of autistic females are a consequence of typical sex differences as they also manifest in typical females during face processing. For example, the main effect of sex found on the P100 amplitude and N170 latencies is consistent with other studies in NA individuals[64–67]. In addition, the difference between HSF MMR and LSF MMR positive peak in the first time window is significant in females but not in males, regardless of group. Although observed in other MMR studies, this central positive activity has rarely been investigated but has been associated with fusiform sources to face stimuli[68] (which is also the source of N170[69]). In line with these findings, higher activity for HSF MMR compared to LSF MMR was found in the first time window in the right fusiform area in both autistic and NA females, but not in males. The results suggest typical fusiform activity during face processing in autistic females. Conversely, in predominantly autistic males samples, atypical fusiform gyrus activity during face processing has been often reported[19] and may be related to spatial frequency processing[70]. Altogether, these findings can be assimilated to a "female protective effect"[71–73]. Since some areas of difficulty in autism are related to areas where females typically have an advantage such as face processing[74], autistic females may indeed be less impaired. This would partly explain why autistic females require a greater genetic load to express similar symptoms to males[75]. From a clinical perspective, the present results in autistic females may be associated with their better attention to faces than autistic males[9–12]. However, future studies are needed to investigate the relationship between attention to faces and prediction error to faces.

On the other hand, findings also indicate that autistic females have peculiarities that are specific to the autistic group. This is supported by the analysis of the second time window showing a reduced difference in activity between HSF MMR and LSF MMR in autism compared to NA, irrespective of sex (see Fig. 2). As late MMR may be associated with deeper processing of faces such as emotion processing[53], this could explain why autistic females still struggle with face processing[13,15,76] and social adaptation compared to NA. The present results suggest that some of these difficulties may be related to predictive coding specificities in autistic females, similar to autistic males. The specific profile of autistic females is also observed in sources. Indeed, in addition to higher activity in the right fusiform for HSF MMR compared to LSF MMR in the first time window (similar to NA females), higher activity was also found in the left fusiform, similar to NA males. In the second time window, only NA females showed higher activity for HSF MMR than LSF MMR. It is located in both hemispheres, in a broad region that includes the lateral occipital cortex. This could align with previous findings in a large database showing higher resting-state connectivity between the two hemispheres in the dorsolateral occipital cortex in NA females compared to NA males, autistic males, and autistic females[77]. In contrast to previous brain structure and connectivity studies showing neural masculinization in autistic females and neural feminization in autistic males[78–80], the present findings of neurophysiological activity during face processing show a mixture of typical males and typical females, as well as autistic males characteristics in autistic females.

Given the present findings, it is advisable to interpret with caution results indicating enhanced camouflaging in autistic females[81–84], and to encourage a more precise construct of camouflaging[85]. Camouflaging in autism is usually described as the use of conscious or unconscious masking or behavioral compensation strategies to minimize the visibility of autism symptoms in social contexts[86]. In the absence of a clear definition of unconscious camouflaging in the literature, we refer here to conscious strategies (those that have been learned/ controlled at

some point). Measures of camouflaging are partially[87] or fully[88] based on self-report questionnaires. If better socio-communicative abilities in autistic females are due to better camouflaging abilities (conscious/controlled processes), autistic males and females should exhibit a similar neurophysiological profile during automatic face processing. However, the divergent brain response observed in autistic males and females appears to be explained by more typical low-level processes (unconscious) in autistic females (although different from those in NA individuals). This discrepancy may result from neural compensatory mechanisms in autistic females, that lead to neurophysiological modulations over time. Alternatively, it may be inherited. In this case, the neurophysiological baseline would differ between autistic males and females, making it inaccurate to draw conclusions about better camouflaging in females. To make such comparisons, a similar baseline or some sort of correction would be required to control for neurophysiological processes. Although the present study sheds light on this critical point, it does not assess whether autistic women camouflage more than men since it lacks a camouflaging measure. Nonetheless, it may be inferred from the fact that females are diagnosed with autism at a later age than males despite having similar AQ scores. Thus, future research should explore how specific camouflaging strategies relate to the neurophysiological response in autistic individuals. Another limitation of the present study is its inability to disentangle inherited and developmental compensatory mechanisms. It emphasizes the critical need for longitudinal studies to better understand developmental trajectories, which remain understudied with respect to sex disparities in autism[89]. These studies would also help to better understand the heterogeneity of the autism spectrum and investigate the existence of distinct subgroups[90,91]. One hypothesis is that the sex ratio may differ according to these subgroups and that the neurophysiological profile observed here in autistic females may be more characteristic of one of the subgroups, as most of the participants were diagnosed late, especially females. This broad area of research still requires a great deal of investigation[92–95].

In conclusion, this study highlights a specific neurophysiological profile in autistic women, which is intermediate between autistic men and NA individuals during the first stages of face processing, and which may contribute to the observable "female autism phenotype"[71]. It suggests that autistic females might be under-recognized due to better social skills than autistic males, but their remaining specificities may explain why they still struggle. It underlines the need to better characterize their profiles in research and urges clinical professionals to be aware of their specificities. Although these specificities remain to be tested with a developmental approach and longitudinal studies, the current findings encourage future research on sex differences in autism to better outline what pertains to camouflaging, compensatory mechanisms, and a female protective effect. Finally, this study also suggests that predictive coding specificities in autism play a role in their difficulties in face processing and adds new empirical data to the literature on predictive coding specificities in autism.

## Methods

**Participants**. A final sample of 41 autistic participants (20 females) and 48 NA participants (24 females) were included in the study. Female and male categories were defined here by biological sex recorded at birth and no participant identified as transgender in the study. All autistic participants were clinically diagnosed by an experienced clinical team, based on the criteria of the Diagnostic and Statistical Manual of Mental Disorders, DSM-IV or DSM-5[4,96]. Because they were recruited primarily through an expert center for adult autism diagnosis, they were often

**Table 1 Mean value, standard deviation and range for age, education (number of school-year since the beginning of elementary school), visual acuity (logMAR), IQ scales and AQ scores as well as the percentage of participants with a diagnosis other than autism for each subgroup, and p-value of group comparison.**

| Variable | Autistic F (N = 20) | Autistic M (N = 21) | TD F (N = 24) | TD M (N = 24) | p value |
|---|---|---|---|---|---|
| Age |  |  |  |  | 0.514 |
| Mean (SD) | 30.9 (8.6) | 27.9 (8.8) | 31.1 (8.6) | 29.5 (5.2) |  |
| Range | 18.4–44.2 | 18.1–46.0 | 19.5–46.1 | 21.2–43.0 |  |
| Education |  |  |  |  | <0.001 |
| Mean (SD) | 14.4 (2.3) | 13.0 (1.5) | 16.3 (1.9) | 15.6 (2.4) |  |
| Range | 11.0–20.0 | 10.0–16.0 | 14.0–20.0 | 11.0–20.0 |  |
| LogMAR |  |  |  |  | 0.977 |
| Mean (SD) | −0.1 (0.1) | −0.1 (0.2) | −0.1 (0.1) | −0.1 (0.1) |  |
| Range | −0.3–0.2 | −1.0–0.2 | −0.3–0.1 | −0.3–0.1 |  |
| FSIQ |  |  |  |  | 0.742 |
| Mean (SD) | 119.1 (13.2) | 116.7 (12.7) | 120.0 (13.1) | 116.9 (9.8) |  |
| Range | 96.0–149.0 | 89.0–136.0 | 92.0–147.0 | 100.0–135.0 |  |
| PIQ |  |  |  |  | 0.825 |
| Mean (SD) | 109.3 (14.3) | 109.9 (14.5) | 109.9 (14.9) | 106.6 (11.3) |  |
| Range | 84.0–136.0 | 73.0–140.0 | 82.0–140.0 | 80.0–134.0 |  |
| VIQ |  |  |  |  | 0.626 |
| Mean (SD) | 126.2 (10.9) | 123.2 (9.8) | 126.2 (11.3) | 123.2 (10.0) |  |
| Range | 100.0–146.0 | 103.0–141.0 | 98.0–147.0 | 100.0–144.0 |  |
| AQ |  |  |  |  | <0.001 |
| Mean (SD) | 36.5 (4.0) | 32.1 (8.7) | 15.0 (8.4) | 17.0 (5.6) |  |
| Range | 30.0–44.0 | 13.0–44.0 | 3.0–32.0 | 9.0–28.0 |  |
| PsyNeuroDiag |  |  |  |  | <0.001 |
| no | 10 (50.0%) | 16 (76.2%) | 23 (95.8%) | 23 (95.8%) |  |
| yes | 10 (50.0%) | 5 (23.8%) | 1 (4.2%) | 1 (4.2%) |  |
| DiagnosticAge |  |  |  |  | 0.013 |
| Mean (SD) | 28.6 (8.6) | 21.3 (9.6) |  |  |  |
| Range | 17.0–42.0 | 5.0–38.0 |  |  |  |

*NA* Non Autistic, *F* females, *M* Males, *FSIQ* Full Scale Intelligence Quotient, *PIQ* Performance Intelligence Quotient, *VIQ* Verbal Intelligence Quotient, *AQ* Autism Quotient.

diagnosed late. 15 reported a co-occurring psychiatric (e.g., anxiety, depression) or neurodevelopmental diagnosis (e.g., dyslexia) and 9 reported taking prescribed medication. Among NA participants, 2 reported an anxiety diagnosis and use of prescribed medication. As this did not affect the performance and EEG data, they were retained in the analyses. Intellectual Quotient (IQ) scores were obtained from the diagnostic records of autistic participants (WAIS-IV, $N = 16$ ; WAIS-III, $N = 1$ ; WISC-IV, $N = 2$). These participants were tested in adulthood or late adolescence (>16 years of age). IQ was estimated for those who could not provide a score of a Weschler test or for those who were tested in childhood ($N = 22$), as well as for NA participants, using four selected subtests of the WAIS-IV (Vocabulary, Similarities, Block Design and Matrix)[97–99]. All participants also completed the Autism Quotient questionnaire, AQ[100]. Seven autistic participants presented an AQ score below 32, which is consistent with the AQ distribution in autism as 80% of autistic individuals score above 32[100], and with its low negative predictive value despite its good sensitivity[101]. The demographics of the final sample are summarized in Table 1. The groups did not differ on age, visual acuity and IQ (all $p > 0.51$). However, according to epidemiologic evidence[102,103], the groups differed in educationnal attainment. Autistic males had lower levels of education than NA males ($p < 0.001$) and autistic females had lower levels of education than NA females ($p = 0.01$) but did not differ from ASD males or NA males. The groups also differed on co-occurring conditions, with autistic females presenting more additional diagnoses than the other groups ($p < .001$), in line with a recent meta-analysis[104]. As expected, autistic participants had higher AQ scores than NA (all $p < 0.001$), but ASD males did not differ from ASD females ($p = 0.20$) and NA males did not differ from NA females ($p = 0.75$). After being informed of the study

objectives and procedures, participants provided written informed consent. The study was approved by the Ethics Committee (Comité de Protection des Personnes Ile de France 1—IRB/IORG: IORG0009918) under agreement number 2019A01145-52. All ethical regulations relevant to human research participants were followed .

**Stimuli and procedure**. Stimuli, procedure and EEG data preprocessing were the same as Lacroix et al.[42]. Th stimuli consisted of two gray-scale photographs of the same actress (Fig. 5a1). They were presented in two sequences. In the oddball sequence (Fig. 5b) of 1575 stimuli, the broad spatial frequencies (BSF) photograph was the standard stimulus and was presented at a probability of $p = 0.80$. The same photographs, filtered in HSF above 6 cycles per degree (dHSF; $p = 0.10$) or in LSF below 1.5 cycles per degree (dLSF; $p = 0.10$) were used as the deviant stimuli. The same BSF photograph but in color was presented as the target ($p = 0.05$, among standards stimuli). Cutoffs of filtered stimuli were chosen in line with previous research and physiological use of SF see ref. [42]. In the equiprobable sequence of 835 stimuli, all stimuli occurred with a probability of $p = 0.16$. This sequence included the same stimuli as in the oddball sequence (eLSF, eHSF and eBSF for equiprobable low, high and broad spatial frequencies respectively) as well as three additional stimuli (eLSF2, eHSF2 and eBSF2) depicting another expression of the same actress. Target stimuli were the same stimuli but colored (Fig. 5a2). Autistic and non autistic participants did not differed on the behavioral response (Supplementary Table 1).

Stimuli were centrally displayed using Presentation® software (Neurobehavioral Systems, Inc., Berkeley, CA, www.neurobs. com) on a CRT screen measuring 37 × 29.6 cm with a refresh rate

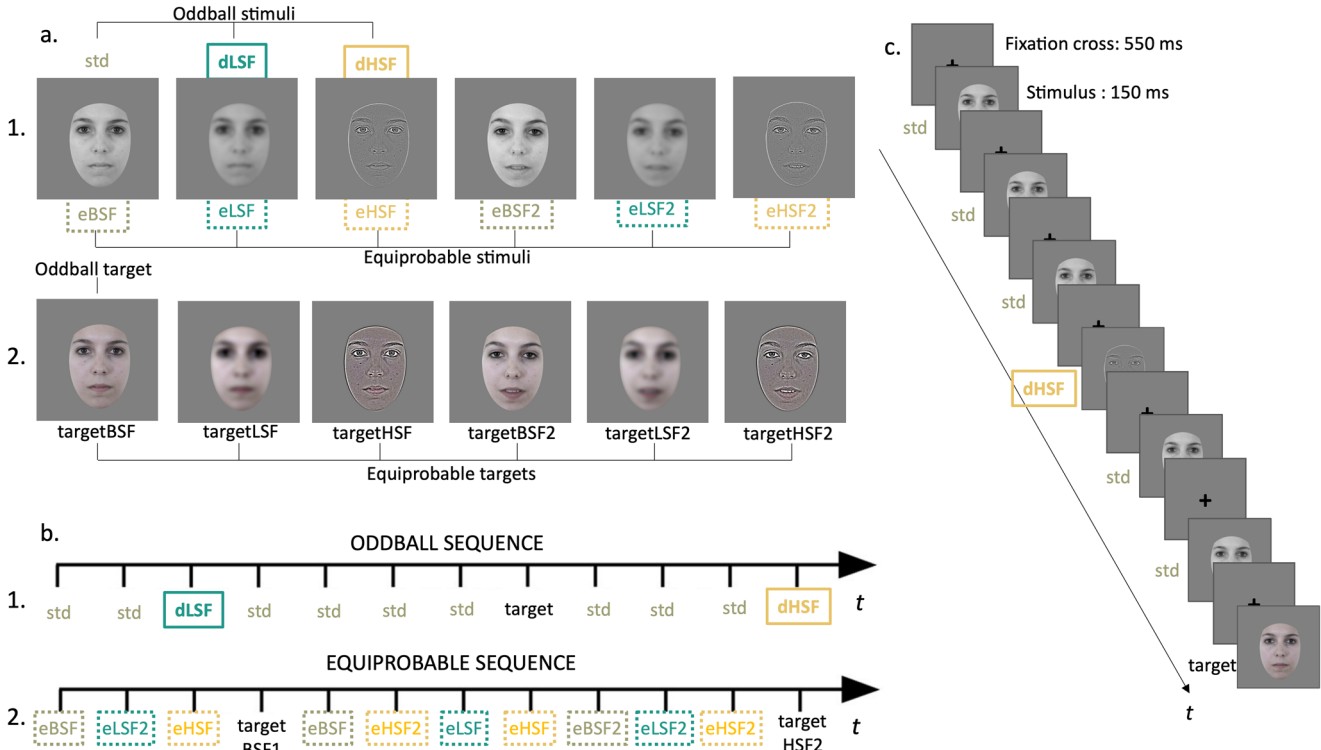

**Fig. 5 Stimuli and procedure. a.1** Gray-scale stimuli presented in the oddball sequences and in the equiprobable sequence. The MMR is calculated as the arithmetic difference between ERPs to deviant in the oddball sequence and ERPs to the same stimulus in the equiprobable sequence, to control for the MMR not being attributed to differences in the physical characteristics of the stimulus. **a.2** Colored targets presented during the oddball sequence and the equiprobable sequence (second line). **b.1** Example of stimuli presentation during the oddball and **b.2** during the equiprobable sequence. **c** Time course of stimuli presentation.

of 75 Hz and a resolution of 1280 × 1024 pixels. They were presented at a viewing distance of 87 cm to maintain a visual angle of 5.8° for each stimulus. The stimuli appeared for 150 ms with a 550 ms inter-stimulus interval during which a fixation cross was presented (Fig. 5c). Participants were instructed to focus on the fixation cross and to promptly detect the colored face by pressing a button as quickly as possible. They were monitored via a camera during the 30-min recording session.

**EEG data**
*EEG recording.* EEG recordings were performed at the IRMaGe neurophysiology facility (Grenoble, France). BrainAmp amplifiers and EasyCaps (Brain Products GmbH, Germany) with 96 active electrodes following the 10–5 standard system were used for EEG recording, with impedance kept below 25 kΩ. A sampling rate of 1000 Hz was used for signal recording, with an anti-aliasing filter at 500 Hz. The ground electrode for the EEG was FPz, while the reference electrode was FCz. Two electrodes on the left and right outer canthi of the eyes and two others above and below the left eye were used to record the horizontal and vertical electro-oculographic (EOG) activity (hEOG and vEOG) respectively. The ground electrode for EOG was positioned on the left base of the neck.

*EEG preprocessing.* Brainstorm software[105] and MATLAB (The MathWorks Inc.) scripts were used for EEG preprocessing. Muscular artifacts were manually discarded for each participant. The signal was then re-referenced using the average reference. Eye movements were corrected using signal space projection (SSP). Finally, a band-pass filter of 0.1–40 Hz was applied to the cleaned signal and trials were epoched from 100 ms pre-stimulus to 600 ms post-stimulus, except for the first three trials of the

sequences and trials presented after the deviant or target, which were excluded. In the end, 1% of the trials from NA participants and 2.3% of the trials from autistic participants were discarded during preprocessing. Next, bad channels were interpolated based on neighboring channels. An average of 4 channels were inter-polated per participant. For six participants, the signal was recorded on only 64 electrodes and missing electrodes (distributed on the scalp) were also interpolated for statistical analyses.

*Event-related potentials.* For each subject and each condition of interest in the oddball sequence (standard, dHSF, dLSF) and in the equiprobable sequence, all trials were averaged. MMRs were computed for HSF and LSF conditions by calculating the arithmetic difference between the ERPs to the deviant in the oddball sequence, and the ERPs to the same stimulus presented in the equiprobable sequence. Finally, grand average waveforms were computed across participants for each ERP and deviant condition.

*Source reconstruction.* The anatomical location of the activity was estimated by source reconstruction using Brainstorm. A realistic forward model based on the ICBM152 template and a standard co-registered set of electrode positions was used. Noise covariance matrices were computed for each participant using the baseline activity of each condition. The source space was restricted to the cortical surface with 2500 dipoles and the inversion kernel was computed using sLoreta[106] assuming SNR 3 and unconstrained orientation. Source reconstruction was performed for each participant in each condition, except for one autistic participant who had an atypical signal in frontal areas. Then, the difference in sources between deviant and equiprobable conditions (for HSF

and LSF) was performed for each participant as well as the difference between HSF MMR and LSF MMR. Finally, the signal was average for each group and subgroup (autistic males, autistic females, NA males and NA females) in each condition.

## Statistics and reproducibility

*ERPs analyses.* The sensory response to unfiltered and filtered stimuli presented in the equiprobable sequence (eBSF, eLSf and eHSF) was analyzed. Peak latencies and amplitudes were extracted using MATLAB scripts (and visual inspection) in the 60–140 ms latency range of for the P100 and 130–200 ms for the N170. As a negative peak was observed for the P100 to HSF filtered stimuli on the most posterior channels (Oz, O1 and O2), P100 analyses were performed over PO7 and PO8. The negative peak for N170 was observed on P7 and P8 and these electrodes were used for N170 analysis.

The parieto-occipital negative activity of the MMR was collected over P7, P8, PO7, PO8, PPO9h, PPO10h, P9 and P10. Central positive activity of the MMR was collected over Cz, CPz, CPP1h and CPP2h. Time windows for analysis were chosen based on previous studies and visual inspection of averaged MMRs in both groups. MMR peak amplitude and latency were analyzed in the first time windows (130–230 ms). As the MMR for LSF appeared to be more sustained in the second time window (350–450 ms), analyses of mean amplitude (rahter than peak amplitude) were performed.

ANOVAs were performed with Group (ASD, NA) and Sex (Female, Male) as between-subject factors and SF (BSF, LSF and HSF for P100 and N170; LSF and HSF for MMR) and Hemisphere (Left, Right) as within-subject factors. Hemisphere was not included as a factor in the analyses of the central positive activity of the MMR, which was measured over a median region. Sphericity was tested with Mauchly tests and Greenhouse–Geisser correction was applied in case of deviation from sphericity. Tukey correction was used for planned comparisons or post-hoc tests.

*Cluster based statistics.* In addition, cluster-based permutation tests (using the ft_timelock statistic, with "Monte-Carlo" and cluster as parameters) were used for MMR analyses. Samples were selected for clustering with a significance threshold of $\alpha = 0.05$. We used dependent paired two-tailed *t*-tests over the 50–600 ms time window after stimulus onset on all electrodes to assess differences between HSF MMR and LSF MMR in each group and sex. We then used independent paired two-tailed *t*-tests over each time window (130–230 ms and 350–450 ms) on all electrodes to assess differences between groups and sex. Significant samples were included in the clustering algorithm with the requirement of at least two neighboring channels. Cluster-level statistics were then calculated by summing the *t*-values within each cluster and the Monte-Carlo procedure (1000 permutations) was used for correction. The significance threshold for clusters was set at $p_{cluster} < 0.05$.

*Source statistics.* Source statistics were performed to analyze differences between-group and differences between HSF MMR and LSF MMR for each subgroup. The norm of each dipole was used for statistical analysis. Cluster-based permutation tests were performed on the mean signal over each time window (130–230 ms and 350–450 ms) using the ft_timelock statistic with the Monte-Carlo procedure (3000 permutations). A correction was applied at the cluster level with a significance threshold for sample selection of $\alpha = 0.05$. The significance threshold for clusters was set to $p_{cluster} < 0.05$. Following this step, the Desikan-Killiany atlas was used to identify significant regions.

**Reporting summary.** Further information on research design is available in the Nature Portfolio Reporting Summary linked to this article.

## Data availability
The ERP data (mean by participant), non-identifying demographic data, and materials used in this study are openly available on the Open Science Framework (OSF) at https://osf.io/n3f2g/?view_only=bcd4c3f36cc74b22b2e26f6bb91ad7ce. The raw data recorded for this study are not publicly available due to absence of public data sharing statement in the informed consent form signed by the participant. The data may be available on request from the corresponding author Adeline Lacroix, on the condition that a data sharing agreement between the academic buyer and Grenoble University Hospital is established.

## Code availability
All statistics were performed using R 4.2.3 and R studio 2023.09.1 + 494. The script, included in the Rmarkdown document, is available on OSF (https://osf.io/n3f2g/?view_only=bcd4c3f36cc74b22b2e26f6bb91ad7ce).

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

## Acknowledgements

A.L. was supported by grants from the Ministry of Higher Education, Research and Innovation (France), the ANR-17-EURE-003 of the CBH graduate school, the IDEX Investissement d'avenir of the EDISCE doctoral school and the GIS autisme et troubles du neurodéveloppement. This work was partially supported by the MIAI @ Grenoble Alpes, (ANR-19-P3IA-0003). The data were acquired on a platform of the France Life Imaging Network, partially funded by grant ANR-11-INBS-0006. We thank all participants for their involvement in the study. We thank the Center Expert Asperger, the C3R of Grenoble, the Center d'Evaluation Savoyard de l'Autisme in Chambéry and the Service Accueil Handicap of the University of Grenoble Alpes for their help in recruiting participants. We would like to thank Valentine Cuisinier, Margot Fombonne and Perrine Porte for their invaluable help with data collection.

## Author contributions

Conceptualization and method: A.L., S.H., M.M., L.B., K.K., M.Go. Funding: A.L., M.M., M.Go.; Data collection: A.L., S.H., and L.V; Data curation: A.L. and S.H; Data analysis: A.L., S.H., and L.B.; Writing original draft: A.L., S.H., M.M., M.Ga., L.B., L.V., D.A., F.D., K.K., and M.Go.; Writing-review & editing: A.L., S.H., M.M., M.Ga., L.B., L.V., D.A., F.D., K.K., and M.Go.

## Competing interests

The authors declare no competing interests.
