## [Peer Review File · Communications Biology]

Reviewers' comments:

Reviewer #1 (Remarks to the Author):

Overall I find this article to be timely and a very important study. The investigation of how the diagnosis of autism can be impacted by sex differences is indeed very critical. The authors have correctly tried to test how these differences are linked to underlying neurophysiological differences. While I find the study design to be very interesting, I think the article needs to be revised thoroughly in the following way.

For the ease of reading and an unbiased interpretation of the results I would suggest the following way to reorganize the results.

First, the difference in the neurophysiological responses between LSF and HSF stimuli during the MMR paradigm should be statistically characterized in NA. Then the difference in these statistics should be quantified in the autistic population (without any difference considered in the sex). All of these should be done purely as neurophysiological signals without any reference to "predictive processes" etc. just as pure empirical results.

Second, a hypothesis that male and female could be same or different in these statistics for autistic adults should be proposed.

Third, the results comparing male and female autistic subjects should be presented purely as neurophysiological results. How this difference compares to the differences in the signal stats reported in NA should be directly contrasted (tested) and reported.

Once the differences are clearly established with statistics — then a speculative discussion should follow (can be supported by figures, further hypotheses development etc.) as to why such differences could exist (based on predictive signals etc.).

In the current version, the link to predictive processes as a neural mechanism explaining the expected difference between male and female from the very beginning makes it very hard to read and interpret for readers who 1) might not be familiar with this literature 2) might require a bit more evidence to tie together EEG signal differences with predictive process based neural hypotheses despite the literature behind it (which sometimes feel more unsubstantiated than presented here).

Reviewer #2 (Remarks to the Author):

The authors report the findings from an EEG study of adults of both sexes with and without autism using an oddball paradigm contrasting static black and white photos of faces in which all the spatial frequencies are present (the standard) versus those in which only HSF or LSF are retained. The authors found evidence suggesting greater mismatch responses to HSF, indicating the usual reliance on LSF in face processing, but that this differential response was reduced in adult males with autism compared to females with autism and compared to typical individuals. The authors conclude that adult females with autism show physiological responses to faces that are less abnormal than those seen in adults with autism.

The authors spend much of the introduction dealing with the concept of camouflaging in females with autism and position their paper in terms of differentiating between camouflaging and neurobiological differences. Nevertheless, the authors use no measure of camouflaging in their subjects and are thus unable to pit one mechanism against another. Additionally, camouflaging typically refers to behaviors, whereas the authors are focusing on sensory processing. My strong suggestion would be to remove the focus on camouflaging in the introduction as it is not addressed experimentally.

There is an extensive literature on face processing in children and adults with ASD using EEG, MEG

and fMRI, and I think it incumbent on the authors to summarise succinctly the previous findings with regard to sex differences.

I think the reader would find it easier to follow if the comments on the predictive brain framework preceded those on MMN.

The analyses appear appropriate, but this section could be shortened to improve impact.

I do have some concerns about the IQ testing. The reported full scale IQ's are high: where were the subjects recruited from? Additionally, the reader would have a better sense of the nature of the sample if it was indicated when the adults received their diagnosis: i.e was this in childhood or in adulthood? It appears that all the typically developing individuals were tested using 4 sub-tests of the WAIS. By contrast, an unknown number of individuals with autism were tested this way, whereas in the other individuals IQ data was extracted from their clinical records. It is unclear whether the extracted data were also obtained using the WAIS, or some other measure. Also, it is unclear whether the IQ in the clinical records was obtained in adulthood or childhood.

Reviewer #3 (Remarks to the Author):

The authors report on sex-specific neurophysiological patterns using a mismatch response visual paradigm. They relate this main finding to camouflaging claims about female autistics and predictive coding specificities in male autistics. The statistical tests are convincing and well-represented in the corresponding figures.

The authors provide evidence for a distinct neurophysiological profile in autistic females during in early processing of human faces. The manuscript contributes novel and interesting pieces of evidence as they relate to gender differences in autism- a field of growing interest and import.

The authors touch upon several 'hot topics' in the introduction and discussion. The work shown here is meaningful in the context of camouflaging, predictive coding, and diagnosis. However, the absence of subheadings or clearer structure made it difficult to easily identify the main question being addressed. The title, which reads too general, does not help identify this easily either. A title like the main result subheading: "Mismatch response specificity in autism is modulated by sex" would provide a lot more context and help identify the goal of the paper clearly.

The following are more detailed points with regard to the introduction:

- The prevalence statement needs more specificity. Is this the worldwide prevalence? Prevalence of population in Norway (citation)?
- A statement like "Indeed, mounting evidence suggests that autistic females exhibit stronger social skills compared to autistic males..." warrants more than just one and a minimum of 3 references. Despite adding systematic review reference.
- At the end of page 6 authors present the aim to help 'distinguish' between 2 hypotheses. This section could benefit from a cleaner breakdown of the two hypotheses- much like how it is presented in the Figure 1 caption.
- Great figure!

The results:

- Last sentence of the first paragraph. What is SF? I am sure it is spatial frequency but not sure that this was spelled out prior.
- Figure 3. Very busy. May be better appreciated as separate A-B and C-D figures.
- "Reduced spatial frequency differentiation in autism is modulated by sex" Would make a great title.

COMMSBIO-23-2282-T

Point by point response to the Reviewers' comments and concerns.

Our comments are in black.

Reviewers' comments are in italic blue.

Manuscript's extracts are in green.

Reviewer #1 (Remarks to the Author):

Overall I find this article to be timely and a very important study. The investigation of how the diagnosis of autism can be impacted by sex differences is indeed very critical. The authors have correctly tried to test how these differences are linked to underlying neurophysiological differences.

Thank you for this positive feedback.

While I find the study design to be very interesting, I think the article needs to be revised thoroughly in the following way.

For the ease of reading and an unbiased interpretation of the results I would suggest the following way to reorganize the results.

First, the difference in the neurophysiological responses between LSF and HSF stimuli during the MMR paradigm should be statistically characterized in NA. Then the difference in these statistics should be quantified in the autistic population (without any difference considered in the sex). All of these should be done purely as neurophysiological signals without any reference to "predictive processes" etc. just as pure empirical results.

We agree with this comment and we appreciated the suggestion. Indeed, in the previous version of the results, we did not clearly emphasize the difference between SF processing for autistic and NA individuals separately. Therefore, as recommended, we have reorganized the results by adding a first paragraph (lines 270-293) entitled "Spatial frequencies modulate the mismatch response differently in autism versus NA", in which we present the difference in MMR between LSF and HSF in NA and then in autism, without considering sex differences. Then, we also present the difference between NA and autism. As suggested, we did not refer to predictive processes here but only to the neurophysiological signal (MMR) :

"In NA, cluster based permutation revealed a reduced difference in LSF MMR (i.e, MMR to deviants containing only LSF) compared to HSF MMR (i.e, MMR to deviants containing only HSF) over centro-parietal areas from 147 to 545 ms ($p_{cluster} = .002$), and over temporo-parietal and occipital regions from 158 to 251 ms ($p_{cluster} = .02$), as well as from 313 to 511 ms ($p_{cluster} = .002$).

In autism, there were two significant clusters indicating a reduced difference in LSF MMR compared to HSF MMR over centro-parietal areas from 108 to 231 ms ($p_{cluster} = .006$), and from 315 to 486 ms ($p_{cluster} = .002$). However, over temporo-parietal and occipital regions, contrary to NA, there was a significant cluster in the first time window only, from 111 to 244 ms ($p_{cluster} = .008$), whereas no significant difference was revealed in the second time window.

This difference between NA and autistic individuals was confirmed with the between group comparison in the two latency ranges of interest, in which cluster-based permutation revealed a reduced difference between HSF MMR and LSF MMR in autism compared to NA between 367 to 411 ms post stimulus ($p_{cluster} = .03$), shown on Figure 3.A)"

Second, a hypothesis that male and female could be same or different in these statistics for autistic adults should be proposed.

This general hypothesis is stated in the introduction (lines 112 – 115), after the theory regarding predictive processes, as it comes from the theory:

“Thus, we hypothesize that the autistic bias toward HSF may contribute to their difficulties in face processing by reducing their predictive processes from LSF. Nevertheless, the better social skills and attention to faces in autistic females compared to autistic males could be explained by more typical predictive processes in autistic females.”

And then, at the end of the introduction (lines 227 -236), the more specific hypothesis about how this difference should translate in the neurophysiological signal is stated. Please note that due to the guidelines of the journal, we have now also added a summary of the results in this paragraph:

« The present study replicates previous findings with a larger sample of NA individuals. In addition, our hypothesis posited that reduced LSF predictions in autistic individuals would result in a smaller discrepancy between their LSF MMR and HSF MMR, relative to the NA group. This hypothesis was confirmed, thus bolstering the validity of predictive brain atypicalities in autism. We also hypothesized that the neurophysiological profile of autistic females would be more typical compared to autistic males, a hypothesis that was supported by the results. The study demonstrated that the neurophysiological response in autistic women during automatic face processing was intermediate between that of autistic men and NA individuals. This may, in part, explain the increased socio-communicative abilities often observed in autistic females compared to males.»

Third, the results comparing male and female autistic subjects should be presented purely as neurophysiological results. How this difference compares to the differences in the signal stats reported in NA should be directly contrasted (tested) and reported.

In accordance with this comment, the new first paragraph of the Results section is followed by results focusing on sex differences by group, addressing this specific hypothesis. We have now reported the results of the interaction effect (SF x Group x Sex) for each evoked potential (MMR1 amplitudes and latency and MMR 2 amplitude, on parieto-occipital and on central areas ; P100 and N170 amplitudes and latencies) to make it clearer. Indeed, in the previous version, the detailed output of each model (including non-significant results) were only available in supplementary materials. After reporting the interaction effect, the planned comparison (contrast) decomposing the interaction between Group, Sex and SF is reported. This decomposition was mainly aimed at testing and quantifying the difference between HSF MMR and LSF MMR between autistic males, autistic females, NA males and NA females. We have now written the detailed results for each subgroup. All changes are highlighted throughout the Results section (lines 294 – 549).

As suggested, we have now changed the writing to present purely the neurophysiological results only, without any reference to predictive coding.

Once the differences are clearly established with statistics — then a speculative discussion should follow (can be supported by figures, further hypotheses development etc.) as to why such differences could exist (based on predictive signals etc.).

In the current version, the link to predictive processes as a neural mechanism explaining the expected difference between male and female from the very beginning makes it very hard to read and interpret for readers who 1) might not be familiar with this literature 2) might require a bit more evidence to tie together EEG signal differences with predictive process based neural hypotheses despite the literature behind it (which sometimes feel more unsubstantiated than presented here).

As recommended, we have now removed the reference to predictive processes from all statistics (lines 294 – 549) to make the results less speculative and the speculative aspects are

developed in the discussion. The links between predictive processes, spatial frequencies and MMR are also explained in the introduction as it is tighten to the method used and to the hypotheses. Nevertheless, we have reorganized the introduction (lines 73 – 216) as suggested by Rewiever 2, for better clarity.

Please also note that we have tried not to overload the result section as the Reviewer 2 recommended to shorten this part. We hope that the addition of statistical results and the removal of the references to predictive processes, following your recommendations, make things clearer now.

Reviewer #2 (Remarks to the Author):

The authors report the findings from an EEG study of adults of both sexes with and without autism using an oddball paradigm contrasting static black and white photos of faces in which all the spatial frequencies are present (the standard) versus those in which only HSF or LSF are retained. The authors found evidence suggesting greater mismatch responses to HSF, indicating the usual reliance on LSF in face processing, but that this differential response was reduced in adult males with autism compared to females with autism and compared to typical individuals. The authors conclude that adult females with autism show physiological responses to faces that are less abnormal than those seen in adults with autism.

The authors spend much of the introduction dealing with the concept of camouflaging in females with autism and position their paper in terms of differentiating between camouflaging and neurobiological differences. Nevertheless, the authors use no measure of camouflaging in their subjects and are thus unable to pit one mechanism against another. Additionally, camouflaging typically refers to behaviors, whereas the authors are focusing on sensory processing. My strong suggestion would be to remove the focus on camouflaging in the introduction as it is not addressed experimentally.

We agree with the reviewer that it would have been interesting to have a camouflaging measure in the study, to verify that camouflaging scores in autistic females are higher than those of autistic males, as usually reported in the literature. Thus, we have modified the limitations section and considered the inclusion of camouflage measures as a perspective for future studies (lines 700 – 705) :

“Although the present study sheds light on this critical point, it does not assess whether autistic women camouflage more than men since it lacks a camouflaging measure. Nonetheless, it may be inferred from the fact that females are diagnosed with autism at a later age than males despite having similar AQ scores. Thus, future research should explore how particular camouflaging strategies relate to the neurophysiological response in autistic individuals.”

Importantly, we followed the strong suggestion of the reviewer and we have removed the reference to camouflaging in the introduction. Accordingly, in the discussion, we have clarified that the aim of the study was to compare the neurophysiological response between autistic males and females and camouflaging is now discussed at the end only (lines 676 – 713).

There is an extensive literature on face processing in children and adults with ASD using EEG, MEG and fMRI, and I think it incumbent on the authors to summarise succinctly the previous findings with regard to sex differences.

We agree with the relevance of adding literature regarding face processing in ASD and we are grateful for this suggestion. We have now added it as follows (lines 75 – 98) :

“Indeed, face processing is an important part of socio-communicative abilities and present qualitative and quantitative differences in autism compared to NA individuals (Tang et al., 2015). Autistic individuals usually recognize emotion from faces less accurately and slower than NA individuals (Harme et al., 2010; Uljarevic & Hamilton, 2013). fMRI shows that the regions recruited during face processing differ between autistic and NA individuals (Campatelli et al., 2013) or that some regions, such as the amygdala are less activated (Costa et al., 2021) or slower (Bathelt et al., 2021). Additionally, a review of EEG studies highlighted a reduced amplitude and a longer latency for the N170 response to face in autism (Monteiro et al., 2017). Despite sex differences in socio-communicational abilities in autism, sex differences in face processing has not been much investigated. One study showed an attenuated response in autistic girls compared to boys (Coffman et al., 2013) but had 24 autistic participants only and no control group. “

I think the reader would find it easier to follow if the comments on the predictive brain framework preceded those on MMN.

Thanks for this relevant advice. We have now rearranged the introduction as recommended (lines 73 – 226), explaining first the face processing peculiarities in autism, then the link with spatial frequency processing and thus the predictive brain framework, and finally, the link with MMR. We hope this make the introduction easier to follow.

The analyses appear appropriate, but this section could be shortened to improve impact.

As Reviewer 1 also suggested, we rewrote this part to make it clearer, implying the report of additional results. Nonetheless, we have also tried to make it shorter by removing the interpretive aspects related to predictive coding. To avoid overloading the Results section, the detailed results (detailed output of the models) are provided in the Supplementary materials.

I do have some concerns about the IQ testing. The reported full scale IQ's are high: where were the subjects recruited from? Additionally, the reader would have a better sense of the nature of the sample if it was indicated when the adults received their diagnosis: i.e was this in childhood or in adulthood?

Participants were mainly recruited through the expert centre for autism diagnostic of Grenoble (France), dedicated to autism in adulthood, but also through organizations for autistic individuals and with the service for disabled students of the university of Grenoble. The main source of recruitment via the expert centre explains why our participants were mainly diagnosed late and we have now mentioned that in the Material and method section (lines 793 – 795) for clarity:

“As they were mainly recruited through an expert centre for autism diagnostic in adulthood, they were often late-diagnosed. »

We have also added the mean age of diagnosis and the range of ages of diagnosis in males and females in Table 1.

It should be noted that higher verbal IQ (which is the profile of our participants) is usually associated with lower severity of autism symptoms (Johnson et al., 2021) and better compensation (Livingston et al., 2018) that could possibly make them unnoticed/late diagnosed. This could explain that our late-diagnosed participants have this type of high IQ profile.

Johnson, C. N., Ramphal, B., Koe, E., Raudales, A., Goldsmith, J., & Margolis, A. E. (2021). Cognitive Correlates of Autism Spectrum Disorder Symptoms. *Autism Research : Official Journal of the International Society for Autism Research*, 14(11), 2405–2411. <https://doi.org/10.1002/aur.2577>

Livingston, L. A., Colvert, E., Social Relationships Study Team, Bolton, P., & Happé, F. (2018). Good social skills despite poor theory of mind: Exploring compensation in autism spectrum disorder. *Journal of Child Psychology and Psychiatry, and Allied Disciplines*. <https://doi.org/10.1111/jcpp.12886>

Although the participants of the study seem to correspond to a specific subgroup of the autistic population, it is important to note that correlational analyses (reported in the Supplementary material) did not show a significant correlation between the age of diagnosis and the mismatch response.

It appears that all the typically developing individuals were tested using 4 sub-tests of the WAIS. By contrast, an unknown number of individuals with autism were tested this way, whereas in the other individuals IQ data was extracted from their clinical records. It is unclear whether the extracted data were also obtained using the WAIS, or some other measure. Also, it is unclear whether the IQ in the clinical records was obtained in adulthood or childhood.

Thank you for your remark. We agree that it was not clear enough and we did the following modification:

“Intellectual Quotient (IQ) scores were collected from the diagnosis record of autistic participants (WAIS-IV, N = 16 ; WAIS-III, N = 1 ; WISC-IV, N = 2). These participants were tested in adulthood or late adolescence (> 16 years old). The IQ was estimated for those who could not provide a score to a Wechsler test or for those who have the test in their childhood (N= 22), as well as for NA participants, using four selected subtests of the WAIS-IV (Vocabulary, Similarities, Block Design and Matrix) (Grégoire & Schmitt, 2021 ; Grégoire & Wierzbicki, 2009; Wechsler, 2008). “

Reviewer #3 (Remarks to the Author):

The authors report on sex-specific neurophysiological patterns using a mismatch response visual paradigm. They relate this main finding to camouflaging claims about female autistics and predictive coding specificities in male autistics. The statistical tests are convincing and well-represented in the corresponding figures.

The authors provide evidence for a distinct neurophysiological profile in autistic females during in early processing of human faces. The manuscript contributes novel and interesting pieces of evidence as they relate to gender differences in autism- a field of growing interest and import.

The authors touch upon several 'hot topics' in the introduction and discussion. The work shown here is meaningful in the context of camouflaging, predictive coding, and diagnosis. However, the absence of subheadings or clearer structure made it difficult to easily identify the main question being addressed. The title, which reads too general, does not help identify this easily either. A title like the main result subheading: "Mismatch response specificity in autism is modulated by sex" would provide a lot more context and help identify the goal of the paper clearly.

Thank you for your interest and positive feedback. Unfortunately, subheadings are not allowed in this journal, except for the result section. We have now reorganized the introduction and hope it is clearer now. We have also changed the title to be more specific with “Sex modulation of faces prediction error in the autistic brain”.

The following are more detailed points with regard to the introduction:

- *The prevalence statement needs more specificity. Is this the worldwide prevalence? Prevalence of population in Norway (citation)?*

As suggested, we added more details and references regarding the prevalence.

“In spite of variation depending on time, geographical location and ethnicity, the prevalence is estimated to be around 1% of the world population (American Psychiatric Association, 2013; Posserud, Skretting Solberg, Engeland, Haavik, & Klungsoyr, 2021; Zeidan et al., 2022). The male-to-female ratio varies from 4:1 in children (Posserud et al., 2021; Roman-Urrestarazu et al., 2021; Zeidan et al., 2022) to 2.6:1 in adults (according to a study based on the medical birth registry of Norway ; Posserud et al., 2021). »

- *A statement like "Indeed, mounting evidence suggests that autistic females exhibit stronger social skills compared to autistic males..." warrants more than just one and a minimum of 3 references. Despite adding systematic review reference.*

Thank you for the suggestion. As recommended, additional references were added :

“Indeed, mounting evidence suggests that autistic females exhibit stronger social skills or more normalized response to social stimuli compared to autistic males (Cauvet et al., 2020; Del Bianco et al., 2022; Harrop et al., 2020, 2018; Harrop et al., 2019; Lacroix, Dutheil, et al., 2022; Sedgewick, Hill, Yates, Pickering, & Pellicano, 2015 Wood-Downie, Wong, Kovshoff, Cortese, & Hadwin, 2021).

- *At the end of page 6 authors present the aim to help 'distinguish' between 2 hypotheses. This section could benefit from a cleaner breakdown of the two hypotheses- much like how it is presented in the Figure 1 caption.*

Thank you for this relevant comment. As strongly recommended by the reviewer 2, the emphasize on camouflaging was removed from the introduction and kept for the discussion. Hence, the section with the 2 hypotheses with the reference to the camouflaging construct was deleted. In the revised version, the general hypothesis is detailed in the introduction (lines 112 – 115), after the theory regarding predictive processes :

“Thus, we hypothesize that the autistic bias toward HSF may contribute to their difficulties in face processing by reducing their predictive processes from LSF. Nevertheless, the better social skills and attention to faces in autistic females compared to autistic males could be explained by more typical predictive processes in autistic females.”

And then, at the end of the introduction (lines 227 – 236), the more specific hypothesis about how this difference should translate in the neurophysiological signal is stated. Please note that due to the guidelines of the journal, we have now also added a summary of the results in this paragraph:

« The present study replicates previous findings with a larger sample of NA individuals. In addition, our hypothesis posited that reduced LSF predictions in autistic individuals would result in a smaller discrepancy between their LSF MMR and HSF MMR, relative to the NA group. This hypothesis was confirmed, thus bolstering the validity of predictive brain atypicalities in autism. We also hypothesized that the neurophysiological profile of autistic females would be more typical compared to autistic males, a hypothesis that was supported by the results. The study demonstrated that the neurophysiological response in autistic women during automatic face processing was intermediate between that of autistic men and NA individuals. This may, in part, explain the increased socio-communicative abilities often observed in autistic females compared to males.»

We hope the aim is clearer now.

- *Great figure!*

Thank you for this positive feedback.

The results:

- Last sentence of the first paragraph. What is SF? I am sure it is spatial frequency but not sure that this was spelled out prior.

Thanks for having noticed that. We have now added the full spelling.

- Figure 3. Very busy. May be better appreciated as separate A-B and C-D figures.

We agree that the figure is busy even if we have tried to make it the clearest possible. However, we think it is important to keep the figure with the 4 panels as they present a similar response (the MMR) and as they are related together (for instance, the dotted lines in B has its correspondences in D; the colors in B and D have their correspondence in the filtered faces in A and allow the reader to have a picture of the MMR without needing to go somewhere else).

- "Reduced spatial frequency differentiation in autism is modulated by sex" Would make a great title.

Thank you for this suggestion. As we also wanted to emphasize that the study was done in electrophysiology and that it concern face processing, we have now changed the title by: "Sex modulation of faces prediction error in the autistic brain". We hope you find it better now.

REVIEWERS' COMMENTS:

Reviewer #1 (Remarks to the Author):

I thank the authors for providing a comprehensive revision.

Reviewer #2 (Remarks to the Author):

I appreciate the comprehensive letter describing the changes made to the manuscript which in my opinion, is much improved. There are a small number of grammatical errors that I believe will be picked up during proof editing, otherwise the MS appears ready for publication.

Reviewer #3 (Remarks to the Author):

The study investigates potential neurophysiological differences between autistic males and females in visual processing. The experimental design using an oddball mismatch negativity (MMN) paradigm to assess spatial frequency processing and predictions during face perception is innovative. The manuscript's readability is much improved and well-written overall after consideration of the reviewer's notes.

I believe the authors have sufficiently addressed my previous notes.

COMMSBIO-23-2282-T

Point by point response to the Reviewers' comments and concerns.

Our comments are in black.

Reviewers' comments are in italic blue.

REVIEWERS' COMMENTS:

Reviewer #1 (Remarks to the Author):

I thank the authors for providing a comprehensive revision.

Thank for acknowledging our efforts in the revision.

Reviewer #2 (Remarks to the Author):

I appreciate the comprehensive letter describing the changes made to the manuscript which in my opinion, is much improved. There are a small number of grammatical errors that I believe will be picked up during proof editing, otherwise the MS appears ready for publication.

We are grateful for your positive assessment. The identified grammatical errors have been addressed and we have indicated the corresponding edits in the manuscript.

Reviewer #3 (Remarks to the Author):

The study investigates potential neurophysiological differences between autistic males and females in visual processing. The experimental design using an oddball mismatch negativity (MMN) paradigm to assess spatial frequency processing and predictions during face perception is innovative. The manuscript's readability is much improved and well-written overall after consideration of the reviewer's notes.

I believe the authors have sufficiently addressed my previous notes.

We are delighted that you find our study innovative and we are grateful for your acknowledgment of the improved readability.